# Adherence of Older Cardiac Patients to a Home-Based Cardiac Rehabilitation Program

**DOI:** 10.3390/geriatrics8030053

**Published:** 2023-05-13

**Authors:** Dennis van Erck, Michel Terbraak, Christine D. Dolman, Peter J. M. Weijs, José P. Henriques, Ronak Delewi, Lotte Verweij, Patricia Jepma, Wilma J. M. Scholte op Reimer, Josje D. Schoufour

**Affiliations:** 1Cardiology, Amsterdam University Medical Centers, University of Amsterdam, 1105AZ Amsterdam, The Netherlands; 2Faculty Health, Center of Expertise Urban Vitality, Amsterdam University of Applied Sciences, 1105BD Amsterdam, The Netherlands; 3Cardiothoracic Surgery, Amsterdam University Medical Centers, University of Amsterdam, 1105AZ Amsterdam, The Netherlands; c.d.dolman@amsterdamumc.nl; 4Faculty of Sports and Nutrition, Center of Expertise Urban Vitality, Amsterdam University of Applied Sciences, 1067SM Amsterdam, The Netherlands; 5Institute for Implementation Science in Health Care, University of Zurich, 8006 Zurich, Switzerland; 6Center of Clinical Nursing Science, University Hospital Zurich, 8400 Zurich, Switzerland; 7Department of Medicine for Older People, Amsterdam University Medical Centers, Vrije Universiteit, 1081HV Amsterdam, The Netherlands; 8Amsterdam Public Health, Aging & Later Life, 1081HV Amsterdam, The Netherlands; 9Research Group Chronic Diseases, HU University of Applied Sciences Utrecht, 3584CH Utrecht, The Netherlands

**Keywords:** home-based cardiac rehabilitation, older cardiac patients, adherence, frailty

## Abstract

Referral to home-based cardiac rehabilitation (HBCR) is low among older and frailer patients due to low expectations regarding adherence by healthcare professionals. The aim of this study was to determine adherence to HBCR when old and frail patients are referred, and to explore any differences in baseline characteristics between adherent and nonadherent patients. Data of the Cardiac Care Bridge were used (Dutch trial register NTR6316). The study included hospitalized cardiac patients ≥ 70 years old and at high risk of functional loss. Adherence to HBCR was confirmed when two-thirds of the intended nine sessions were followed. Of the 153 patients included (age: 82 ± 6 years, 54% female), 29% could not be referred due to death before referral, not returning home, or practical problems. Of the 109 patients who were referred, 67% adhered. Characteristics associated with non-adherence were older age (84 ± 6 vs. 82 ± 6, *p* = 0.05) and higher handgrip strength in men (33 ± 8 vs. 25 ± 11, *p* = 0.01). There was no difference in comorbidity, symptoms, or physical capacity. Based on these observations, most older cardiac patients who return home after hospital admission appear to adhere to HBCR after referral, suggesting that most older cardiac patients are motivated and capable of receiving HBCR.

## 1. Introduction

Cardiac rehabilitation programs after hospital dismissal are effective and commonly available as usual care [1,2]. Participation in cardiac rehabilitation can reverse or prevent functional decline, can increase quality of life, and can reduce the risk of readmission and early mortality [3]. Despite the evidence of their effectiveness, these programs are widely underutilized, with referral rates around 50% in several Western countries [4,5]. Healthcare professionals often cite low expectations regarding patient ability and motivation to adhere as the main reason for not referring patients to rehabilitation programs, particularly in older and more frail patients [2,4,6]. To address this issue, home-based cardiac rehabilitation (HBCR) programs have been developed. It is known that HBCR has the same effect as center-based approaches [7]. However, for specific groups of patients, such as older and frailer patients, who are unable to travel to the hospital, HBCR can remove barriers to participate [8]. However, it is still unknown whether these older and frailer patients adhere to HBCR after referral [9]. The aim of this study is to determine the adherence rates to HBCR in frail and older cardiac patient after referral. In addition, we aim to explore any differences in patient characteristics between referred versus non-referred and adherent versus non-adherent patients. 

## 2. Materials and Methods

### 2.1. Study Design

The data of the intervention group of the Cardiac Care Bridge randomized clinical trial were used in this study (n = 153). Further information about the complete trial is provided elsewhere [10,11]. In brief, the Cardiac Care Bridge is a multicenter randomized clinical trial in the Netherlands that was conducted between June 2017 and March 2020. The aim of the study was to determine the effectiveness of an in-hospital geriatric assessment-based care plan followed by specialized community nurse care in combination with HBCR in reducing six-month readmission and mortality rates. The intervention consisted of three main components: case management, disease management, and HBCR. The study was approved by the local Medical Ethics Committee (Protocol ID: MEC2016_024) and registered in the Dutch Trial Register (https://clinicaltrialregister.nl/nl/trial/24273 NTR6316, 6 April 2017, accessed on 1 February 2023). All patients provided informed consent prior to enrollment, and the study was conducted in accordance with the Declaration of Helsinki. 

### 2.2. Participants

Included participants were cardiac patients, aged ≥70 year and admitted for ≥48 h to one of the participating departments of cardiology or cardiothoracic surgery. Furthermore, patients had to be at high risk of functional loss according to the Dutch Safety Management System (DSMS ≥ 2 in patients aged 70–79 and ≥1 in patients aged ≥80 years) or with a hospital admission in the prior six months. Exclusion criteria included (1) cognitive impairment (MMSE < 15) or delirium, (2) congenital heart disease, (3) life expectancy ≤ 3 months, (4) planned discharge to a nursing home, (5) discharge to a non-participating hospital, and (6) inability to communicate in Dutch. 

### 2.3. Intervention and Home-Based Cardiac Rehabilitation

Patients in the intervention group received a comprehensive geriatric assessment-based integrated care plan, a face-to-face handover to the community care registered nurse and four home visits by this nurse. The pharmacist was also involved, providing advice on medication management. Additionally, a HBCR program was offered to all patients, which involved a physical therapist visiting them at their home. The program consisted of nine sessions over a period of six weeks, in accordance with the Dutch cardiac rehabilitation guidelines [12]. The HBCR program was provided by a trained primary care physical therapist and consisted of a stepwise graded exercise approach, starting with low-intensity functional rehabilitation tailored to patients’ personal goals. The physiotherapists kept a logbook of patients and made notes about their progress and adherence to the program. 

### 2.4. Measures

To describe the patient group and determine the difference between adherent and nonadherent patients, a comprehensive list of collected baseline characteristics was collected by a research nurse. General demographic information included age, sex, educational level, living arrangement and socioeconomic status (calculated from the patients’ postal code of residence by the Netherlands Institute for Social Research (SCP), based on income, employment, and educational level). Hospitalization characteristics included acute or elective hospitalization and previous hospital admission in the past six months. Lifestyle factors included body mass index (BMI) and smoking status. Other conditions that were assessed included nutrition status (short nutritional assessment questionnaire; SNAQ) [13], fall risk (six months fall history), fear of falling and fatigue (numeric rating scale; NRS), cognitive functioning (mini-mental state examination; MMSE) [14], comorbidities (Charlson comorbidity index; CCI) [15], depression (geriatric depression scale; GDS) [16], anxiety (hospital anxiety and depression scale; HADS) [17], self-reported dyspnea and dizziness, number of medications, physical performance (short physical performance battery; SPPB) [18], activities of daily living (the Amsterdam linear disability scale; ALDS) [19], and muscle strength (handgrip strength using a handheld dynamometer) [20]. 

### 2.5. Data Analysis 

Adherence to HBCR was determined using the logbook of the physiotherapist. First, patients were divided into two groups: those who were able to receive a referral to HBCR and those who were unable to do so (e.g., due to mortality or eventually not returning home). Next, the patients who were able to start HBCR were divided into an adherent and non-adherent group. Adherence was defined as participating in at least two-thirds of the intended nine sessions (which corresponds to at least one session per week), or participating in intended sessions until termination due to external, non-motivation-related reasons (e.g., mortality, readmission, practical problems). Patients with early termination due to external reasons were assigned to the adherence group, because HBCR was delivered as intended for these patients. Patient characteristics were then compared between the two groups that were able to receive referral versus those unable, and between the adherent and non-adherent group. Patient characteristics are shown as mean and standard deviation, median and interquartile range, or number and percentage. Differences were estimated using unpaired sample *t*-tests, Wilcoxon signed-rank tests, chi-square tests, or Fischer exact tests depending on the distribution. All analyses were performed in R version 4.2.0. 

## 3. Results

In total, 153 patients were included in the intervention group of the Cardiac Care Bridge (Figure 1). Of these, 29% (n = 44) of the patients were not referred to HBCR. Reasons for non-referral were death before referral (n = 15), not returning home (n = 25), or practical problems (n = 3). Of the 109 patients referred to HBCR, 67% (n = 72) were adherent and 33% (n = 37) were non-adherent to HBCR. The median number of rehabilitation sessions attended by patients in the adherence group was 7.5 [IQR 5.0–9.0]. Among patients in the non-adherence group, 68% (n = 25) did not participate in any rehabilitation sessions, while 32% (n = 12) started but dropped out before achieving two-third of the intended nine sessions. The reasons for non-adherence were no motivation (n = 26), already receiving physiotherapy with a non-cardiac focus (n = 6), or unknown reasons (n = 5). In the adherent group, 54 of 72 patients (75%) completed the full program and 18 patients (25%) were early terminated due to external reasons such as death (n = 5), readmission (n = 6), physical inability to continue as indicated by the physiotherapist (n = 5), or practical problems (n = 2).

Patient characteristics of the different groups are described in Table 1. The patients not receiving a referral to HBCR had lower levels of ADL (median 64 [IQR 54–78] vs. 75 [IQR 60–86], *p* < 0.01), higher geriatric depression score (median 3 [IQR 2–6] vs. 3 [IQR 2–4], *p* = 0.03), higher experienced fatigue (median 6 [IQR 4–7] vs. 5 [IQR 3–6], *p* = 0.03), lower SPPB (median 3 [IQR 1–5] vs. 5 [IQR 3–7], *p* = 0.01), and lower handgrip strength in males (mean 20 ± 11 vs. 27 ± 10, *p* = 0.02). For the adherent versus non-adherent group, no difference was found for most baseline characteristics, including comorbidities or physical functioning. Only two significantly different characteristics were found: age (mean 84 ± 6 in non-adherent vs. 82 ± 6 in adherent group, *p* = 0.05) and handgrip strength in men (mean 33 ± 8 in non-adherent vs. 25 ± 11 in the adherent group, *p* = 0.01). 

## 4. Discussion

We observed that the majority (67%) of old and frail cardiac patients adhered to HBCR after referral, and 75% of these patients fully completed all HBCR sessions. Non-adherent patients were slightly older and men had stronger handgrip strength. No significant differences in other characteristics, including comorbidity, symptoms, or physical capacity, were seen between adherent and non-adherent patients.

In the Cardiac Care Bridge study, 29% of the patients were unable to receive a referral to HBCR, mainly due to institutionalization or death before HBCR could be started. These patients who did not receive a referral to HBCR had lower physical functioning levels, more depressive symptoms, and higher self-reported fatigue at baseline compared to patients who could start HBCR. This is consistent with previous research, including a systematic review and a large cohort study, which have found that lower physical functioning is a significant predictor of institutionalization [21,22]. Depression, fatigue, and low functioning have been identified as predictors of early mortality in older adults [23]. While it was our aim to refer all patients to HBCR, it should be acknowledged that this was not possible for a significant portion of the older and frail cardiac population.

Our data shows that a majority of the older patients referred to HBCR are adherent. An earlier overview of studies showed that approximately 70% of patients around age 60 are adherent after referral to HBCR [7]. This adherence rate is similar to the 67% we found in our older and frailer patient population. This suggests that age does not significantly affect adherence. However, in our data, we did observe a small but statistically significant difference of two years in age between the adherent and non-adherent patients. One possible explanation for this difference could be that the oldest patients live beyond their self-estimated life expectancy, which may reduce their motivation to participate in HBCR [24]. This idea is supported by a qualitative study in our patient group, which showed that patients often questioned the benefits of lifestyle modifications at older ages [25]. In addition to age, we found that male participants in the non-adherent group had somewhat higher handgrip strength than males in the adherent group. It is possible that males with higher strength perceive less threat from their hospital admission, which is known to influence motivation to start a rehabilitation program [25]. Although higher muscle strength does have a better prognosis, rehabilitation is still useful for maintaining strength [26]. Healthcare providers should address false beliefs about the low effectiveness of cardiac rehabilitation in the oldest patients and those with already higher strength. 

Our study has several recommendations for clinical practice and future research. With our high adherence rate, similar to younger patients, we show that even the older and frailer cardiac patients, after accepting referral, are motivated to adhere to HBCR. Currently, referral rates to cardiac rehabilitation by healthcare professionals are low (around 50%), with even lower rates in patients with higher age and frailty [4]. Low referral by healthcare professionals is mainly caused by low expectations about the motivation and capability of the patient to perform a rehabilitation program [2]. Our study shows that these assumptions are unfounded, as most patients were capable and motivated to participate. However, as adherent patients in our study dropped out due to mortality and physical inability, future research should determine for which older and frail patients HBCR is most effective. In addition, our intervention showed no effect on the main outcomes of mortality and readmission [10]. More research is needed to evaluate the effectiveness of HBCR in older and frailer patients. This includes investigating the appropriate content and intensity of the program as well as the potential benefits of incorporating nutritional support. Previous research has shown that a combination of supplemental protein nutrition and exercise is more effective than exercise alone [27,28]. 

There are several limitations to our study. First, HBCR was part of a larger intervention, which may have influenced participation rates. Because of the intensive additional care, patients may have been less motivated to adhere to HBCR. On the other hand, this additional care could also have increased adherence rates. Second, not all older cardiac patients who could benefit from HBCR were included in the Cardiac Care Bridge program, due to strict exclusion criteria [10]. Furthermore, patients with low physical performance were more likely in the group of patients that were not referred to the program as they were not discharged to home, and, therefore, home-based cardiac rehabilitation could not be initiated. However, these patients are the ones who would most likely benefit the most from cardiac rehabilitation [29]. Despite the strict exclusion criteria, we were still able to include a truly vulnerable older population, in which good adherence was still found. Third, our sample size was small, which only allowed for exploratory analysis. Larger studies are needed to confirm our findings. Fourth, the threshold of two-thirds of the intended sessions for adherence is arbitrary. Complete adherence is most likely the most beneficial, as previous research has shown that the number of attended sessions is predictive of the change in exercise capacity [30]. Finally, qualitative data on why patients were not motivated to participate in HBCR were limited. More in-depth interviews would be needed to study this issue.

## 5. Conclusions

Our study found that the majority of older cardiac patients adhered to HBCR after referral. There were no significant differences in comorbidity, symptoms, or physical capacity between the adherent and non-adherent groups. However, we did find that older patients and men with higher handgrip strength were more likely to be non-adherent. Further research is needed to understand why these patients did not participate in HBCR. Overall, our findings suggest that most older cardiac patients are motivated and capable of receiving HBCR.

## Figures and Tables

**Figure 1 geriatrics-08-00053-f001:**
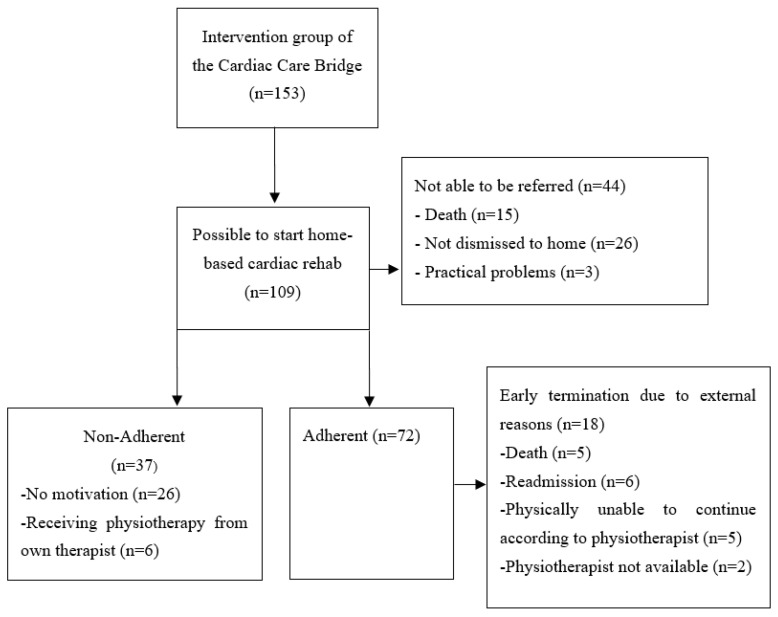
Flow diagram.

**Table 1 geriatrics-08-00053-t001:** Baseline characteristics.

		Unable to Initiate (n = 44)	Able to Initiate (n = 109)	*p*	Able to Initiate (n = 109)		
					Non-Adherence (n = 37)	Adherence (n = 72)	*p*
Demographics							
Age	Years, Mean (SD)	83 (6)	82 (6)	0.91	84 (6)	82 (6)	0.05 *
Sex	Male	45	46	>0.99	46	46	>0.99
Education	Primary Education	39	45	0.52	40	47	0.56
	Secondary Education	41	31		38	28	
	Higher Education	20	24		22	25	
Living arrangement	Living together	41	44	0.86	38	47	0.46
Socioeconomic status	Low (<1 SD)	23	17	0.52	14	20	0.56
	Intermediate	59	65		67	62	
	High (>1 SD)	18	18		19	18	
Body mass index	kg/m^2^, Mean (SD)	27 (6)	27 (6)	0.73	27 (6)	27 (6)	0.84
Current smoker		7	12	0.51	11	11	>0.99
Medical							
Hospitalisation	Acute	98	88	0.12	92	86	0.57
Diagnosis	Heart failure	68	51	0.25	46	54	0.54
	Rhythm or conduction disorder	16	18		16	19	
	Acute coronary syndrome	9	14		22	10	
	Valve deficit	7	10		11	10	
	Other	0	6		5	7	
Previous Hospital admission	<6 months before index event	36	46	0.37	43	47	0.85
Malnutrition	SNAQ	48	30	0.06	32	29	0.90
Fall risk	fall < 6 months	43	44	>0.99	35	49	0.26
Fear of falling	NRS > 4	52	37	0.11	35	38	0.97
Fatigue	NRS, Median (IQR]	6 (4–7]	5 (3–6]	0.03 *	5 (3–6]	5 (4–7]	0.53
Comorbidities	Charlson, Median (IQR]	3 (1–4]	2 (1–4]	0.63	2 (1–3]	3 (1–4]	0.37
Dyspnoea	Self-reported	86	80	0.47	70	85	0.13
Dizziness	Self-reported	41	43	0.94	32	49	0.16
Polypharmacy	>5	82	82	0.87	83	82	0.76
Psychosocial							
Cognitive impairment	MMSE 15–23	6	5	0.28	5	5	0.33
Depression	GDS, Median (IQR]	3 (2–6]	3 (2–4]	0.03 *	3 (2–3]	3 (2–4]	0.24
Anxiety	HADS-A, Median (IQR]	4 (2–7]	3 (1–5]	0.18	2 (1–4]	3 (2–5]	0.32
Physical							
Physical performance	SPPB, Median (IQR]	3 (1–5]	5 (3–7]	0.01 *	5 (3–8]	5 (3–7]	0.36
ADL-functioning	ALDS-score (0–100) ^†^	64 (54–78]	75 (60–86]	<0.01 *	72 (64–83]	78 (60–87]	0.77
Handgrip strength	Male (kg), Mean (SD)	20 (11)	27 (10)	0.02 *	33 (8)	25 (11)	0.01 *
	Female (kg), Mean (SD)	16 (5)	17 (6)	0.40	16 (5)	18 (6)	0.11

* Statistically significant *p* < 0.05. abbreviations: ADL: Activities of daily living, SNAQ: Short Nutritional Assessment Questionnaire, NRS: numeric rating scale, MMSE: Mini-mental state examination, GDS: Geriatric depression scale, HADS: Hospital Anxiety and Depression Scale, SPPB: Short Physical Performance Battery. ^†^ a higher score on ALDS indicates a better performance.

## Data Availability

Data is available on figshare repository https://doi.org/10.21943/auas.14465316.v1, accessed on 1 June 2022.

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
