# Peer review of "Adherence of Older Cardiac Patients to a Home-Based Cardiac Rehabilitation Program"

_geriatrics, 2023, doi:10.3390/geriatrics8030053_

Round 1

Reviewer 1 Report

Thank you for your important work of investigating adherence to home-based cardiac rehabilitation in elderly patients.  I have some suggestions to potentially improve the manuscript.

General comments:

1. I would suggest adding the following variable:  Number of cardiac rehabilitation sessions attended for the referred patients.  Did all of the referred patients participate in at least 1 session?

2. How was the home-based cardiac rehabilitation program accomplished by the physical therapist?  Home visits?  Virtual visits?

Specific comments:

1. I would suggest using a more recent systematic review for reference 3. [Dibben et al.  Exercise based cardiac rehabilitation for coronary heart disease. Cochrane Database Syst Rev 2021 Nov 6: 11 (11):cd001800--benefits include reduced risk of MI, decrease all-cause mortality, reduced hospitalizations, improved health-related quality of life]

2. Line 39: change "admission" to "dismissal"

3. Lines 66-67: Sentence is unclear, I'm not sure what you are trying to communicate.

4. Table 1: Add CV diagnoses: MI, PCI, CABG, CHF, etc.

5. Figure 1: The legend is not correct. The box "Not able to start" Do you mean "Not able to be referred"? The line "Not referred to home" should be changed to "Not dismissed to home".

6. Lines 217-218: Consider changing the phrase "...even the older and frailer cardiac patients are motivated to adhere to HBCR” to " ...even the older and frailer cardiac patients, after accepting referral, are motivated to adhere to HBCR”.

Minor editing needed, as noted above.

Author Response

Thank you for your important work of investigating adherence to home-based cardiac rehabilitation in elderly patients.  I have some suggestions to potentially improve the manuscript.

We thank the reviewer for taking the time to carefully review our manuscript and for providing valuable feedback. We appreciate the insightful comments and suggestions, which have helped us to improve the quality and clarity of our work. Find a point by point reaction on the provided feedback below.

General comments:

  1. I would suggest adding the following variable:  Number of cardiac rehabilitation sessions attended for the referred patients.  Did all of the referred patients participate in at least 1 session?

We thank the reviewer for the suggestion and have now included the number of cardiac rehabilitation sessions attended by the referred patients in the result section. See page 3.

“The median number of rehabilitation sessions attended by patients in the adherence group was 7.5 [IQR 5.0 - 9.0]. Among patients in the non-adherence group, 68% (n=25) did not participate in any rehabilitation sessions, while 32% (n=12) started but dropped out before achieving two-third of the intended nine sessions.”

  1. How was the home-based cardiac rehabilitation program accomplished by the physical therapist?  Home visits?  Virtual visits?

We appreciate this valuable question and have included a sentence in the methods section to clarify that the home-based cardiac rehabilitation program was accomplished by physical therapists through home visits. See page 2.

Additionally, a HBCR program was offered to all patients, which involved a physical therapist visiting them at their home.”

Specific comments:

  1. I would suggest using a more recent systematic review for reference 3. [Dibben et al.  Exercise based cardiac rehabilitation for coronary heart disease. Cochrane Database Syst Rev 2021 Nov 6: 11 (11):cd001800--benefits include reduced risk of MI, decrease all-cause mortality, reduced hospitalizations, improved health-related quality of life]

We thank the reviewer for the suggestion and have updated our reference 3 to the more recent systematic review of Dibben et al. (2021).

  1. Line 39: change "admission" to "dismissal"

We have changed "admission" to "dismissal" on line 39.

  1. Lines 66-67: Sentence is unclear, I'm not sure what you are trying to communicate.

Thank you for bringing the unclear sentence on lines 66-67 to our attention. Upon further consideration, we have decided to remove this sentence from the manuscript as it did not contribute to understanding of our study design and could potentially cause confusion for readers.

  1. Table 1: Add CV diagnoses: MI, PCI, CABG, CHF, etc.

We thank the reviewer for the suggestion and have added CV diagnoses to table 1.

  1. Figure 1: The legend is not correct. The box "Not able to start" Do you mean "Not able to be referred"? The line "Not referred to home" should be changed to "Not dismissed to home".

We appreciate the reviewer's attention to detail and have made the following changes to Figure 1: (a) we have changed the box "Not able to start" to " Not able to be referred" (b) we have changed the line "Not referred to home" to " Not dismissed to home" (c) we corrected the title of the figure. 

  1. Lines 217-218: Consider changing the phrase "...even the older and frailer cardiac patients are motivated to adhere to HBCR” to " ...even the older and frailer cardiac patients, after accepting referral, are motivated to adhere to HBCR”.

We  have revised the phrase as recommended.

Reviewer 2 Report

The article has merit for its theme and methodological conduct.

I suggest removing from the analysis of Table 1 the patients who were unable to initiate the protocol and only leaving those who were able to initiate, maintaining this comparison of the 109 patients.

Thus, reducing confounding biases.

Minor editing of English language required.

Author Response

Thank you for taking the time to provide feedback on our article. We appreciate your suggestion and would like to further explain our reasoning for including patients who were unable to initiate the protocol in Table 1.

Initially, our aim was to refer all older and frail cardiac patients to home-based cardiac rehabilitation (HBCR). However, we were unable to refer all patients, as some were not returning home. Reporting only the patients who were able to initiate the protocol would not provide a complete picture of the referral process for older frailer cardiac patients to HBCR in the clinical setting.

By including patients who were unable to initiate in Table 1, we aimed to also provide insight into the patients that were not able to participate in HBCR. We believe this approach provides the most complete overview of adherence to HBCR in older and frailer cardiac patients and allows readers to make a more informed interpretation of the data and how the clinical practice works.

In summary, while we acknowledge your suggestion of removing patients who were unable to initiate the protocol from Table 1, we believe that including them provides a more comprehensive picture of the referral process for older vulnerable patients to HBCR. Thank you again for your feedback, and we hope that our explanation clarifies the rationale behind our approach.

Reviewer 3 Report

The authors provide data about adherence to a home-based CR program regarding older cardiac patients.

- Introduction: a detailed interpretation of pubilshed benefits of home-based CR ist missing. If there is no benefit compared to classical inpatient or outpatient programs, this should be mentioned.

- How do the authors explain the high percentage of women participating in this program. In most other CR studies, women range from 25-30%.

- The authors decided to accept adherence upon 2/3 participation. Of these patients only 50%  fully competed the short CR program. Previous data from the US and recent publication from Bierbauer et al /EJPC 2018 show, that benefits depend on the number of participation

- a limitation is, that patients with  lower ADLs are not or to a lesser extend participating, although these are the ones who would benefit most. This should be commented

ok

Author Response

The authors provide data about adherence to a home-based CR program regarding older cardiac patients.

Thank you for taking the time to review the paper on adherence to a home-based CR program for older cardiac patients. Your feedback is greatly appreciated and was helpful in improving the quality of the research.

- Introduction: a detailed interpretation of pubilshed benefits of home-based CR ist missing. If there is no benefit compared to classical inpatient or outpatient programs, this should be mentioned.

We agree that a more detailed interpretation of published benefits of home-based CR versus center based CR is helpful. We have revised the introduction, line 46 to 50.

“To address this issue, home-based cardiac rehabilitation (HBCR) programs have been developed. It is known that HBCR has the same effect as center-based approaches [7]. However, for specific groups of patients, such as older and frailer patients, who are unable to travel to the hospital, HBCR can remove barriers to participate [8].”

- How do the authors explain the high percentage of women participating in this program. In most other CR studies, women range from 25-30%.

The majority of studies on cardiac rehabilitation (CR) tend to focus on patients in the age range of 50 to 70 years old. It is a fact that heart problems tend to affect more males in this age group. In the case of patients over the age of 80 years old, it is worth noting that a significant portion of the male population may have already passed away, resulting in a patient group that mainly consists of women.

- The authors decided to accept adherence upon 2/3 participation. Of these patients only 50%  fully competed the short CR program. Previous data from the US and recent publication from Bierbauer et al /EJPC 2018 show, that benefits depend on the number of participation.

Thank you for this comment. In terms of our decision to accept adherence upon 2/3 participation, we agree that it is more beneficial to follow the complete program of nine sessions. We have included a sentence in the discussion session referring to the limitation that 2/3 of the sessions is an arbitrary threshold and that complete adherence is most likely the most beneficial. See line 242 tot 245.

“Fourth, the threshold of two-thirds of the intended sessions for adherence is arbitrary. Complete adherence is most likely the most beneficial, as previous research has shown that the number of attended sessions is predictive of the change in exercise capacity [30].”

- a limitation is, that patients with  lower ADLs are not or to a lesser extend participating, although these are the ones who would benefit most. This should be commented

We agree that the limitation of lower ADL patients not participating in the program should be commented on. We revised the discussion section, see line 234 – 240.

Furthermore, patients with low physical performance were more likely in the group of patients that were not referred to the program as they were not discharged to home, and therefore, home-based cardiac rehabilitation could not be initiated. However, these patients are the ones who would most likely benefit the most from cardiac rehabilitation [29].”

Round 2

Reviewer 3 Report

The current version of the manuscript improved significantly.

Order/total number of references is flawed (30 vs 29 references)

Author Response

Thank you for providing feedback on the current version of the manuscript. We appreciate your comment regarding the number of references in the text. Upon checking our reference software, we discovered that an error had occurred, resulting in a discrepancy between the total number of references and the number of references mentioned in the text.

We have now corrected this mistake and ensured that the references are correct. Additionally, we want to thank you for your positive feedback on the improvements we have made to the manuscript.